# Prognostic Factors of Severe Fever with Thrombocytopenia Syndrome in South Korea

**DOI:** 10.3390/v13010010

**Published:** 2020-12-23

**Authors:** Misun Kim, Sang Taek Heo, Hyunjoo Oh, Suhyun Oh, Keun Hwa Lee, Jeong Rae Yoo

**Affiliations:** 1Department of Internal Medicine, Jeju National University Hospital, Jeju 63241, Korea; drkms1016@gmail.com (M.K.); neosangtaek@naver.com (S.T.H.); hjoh27tr@gmail.com (H.O.); syunni14@naver.com (S.O.); 2Department of Internal Medicine, College of Medicine and Graduate School of Medicine, Jeju National University, Jeju 63241, Korea; 3Department of Microbiology, College of Medicine, Hanyang University, Seoul 04763, Korea; yomust7@gmail.com

**Keywords:** severe fever with thrombocytopenia syndrome, ambient temperature, clinical characteristics

## Abstract

Severe fever with thrombocytopenia syndrome (SFTS), a tick-borne infectious disease, is difficult to differentiate from other common febrile diseases. Clinically distinctive features and climate variates associated with tick growth can be useful predictors for SFTS. This retrospective study (2013–2019) demonstrated the role of climatic factors as predictors of SFTS and developed a clinical scoring system for SFTS using climate variables and clinical characteristics. The presence of the SFTS virus was confirmed using reverse transcription polymerase chain reaction (RT-PCR) tests. In the univariate analysis, the SFTS-positive group was significantly associated with higher mean ambient temperature and humidity compared with the SFTS-negative group (22.5 °C vs. 18.9 °C; 77.9% vs. 70.7%, all *p* < 0.001). In the multivariate analysis, poor oral intake (Odds ratio [OR] 5.87, 95% CI: 2.42–8.25), lymphadenopathy (OR 7.20, 95% CI: 6.24–11.76), mean ambient temperature ≥ 20 °C (OR 4.62, 95% CI: 1.46–10.28), absolute neutrophil count ≤ 2000 cells/μL (OR 8.95, 95% CI: 2.30–21.25), C-reactive protein level ≤ 1.2 mg/dL (OR 6.42, 95% CI: 4.02–24.21), and creatinine kinase level ≥ 200 IU/L (OR 5.94, 95% CI: 1.42–24.92) were significantly associated with the SFTS-positive group. This study presents the risk factors, including ambient temperature and clinical characteristics, that physicians should consider when suspecting SFTS.

## 1. Introduction

Severe fever with thrombocytopenia syndrome (SFTS) is a tick-borne infectious disease that occurs in China, South Korea, and Japan [1,2,3]. It is associated with the genus *Banyangvirus* from the family *Phenuiviridae* of the order *Bunyavirales*, transmitted by either tick bites or close contact with infected patients [1,3]. The incidence of SFTS has been increasing. The cumulative number of cases reported in South Korea by June 2020 was 1125 [4]; in China by 2018, 11,995; and in Japan by March 2019, 402 [5,6,7]. Mortality rate estimates range between 5–30% in East Asia [4].

Clinical manifestations of SFTS include fever, headache, myalgia, and gastrointestinal symptoms, as well as non-specific symptoms. Due to a current lack of effective treatment, early diagnosis is key to adequate supportive care and the prevention of secondary transmission [1,8]. Among patients with a history of insect bites, SFTS is indistinguishable from other diseases such as tsutsugamushi, murine typhus, or anaplasmosis. Since laboratory findings are non-specific, real-time reverse transcription polymerase chain reaction (RT-PCR) tests have been used for the early diagnosis of SFTS [9].

In South Korea, requests for RT-PCR tests increased from 2876 in 2017 to 3391 in 2018 (an increase of 17.9%) [10]. Among these tests, SFTS-positive findings were only confirmed in 272 (9.7%) and 259 (7.6%) cases in 2017 and 2018, respectively. The small percentage of confirmed tests compared to the total number of tests suggested that patients should be assessed for SFTS risk based on other factors, including their clinical characteristics and climatic features of their area of residence. Some studies compared SFTS and scrub typhus [11,12]; one of these studies explored the prediction score generated from three variables, namely leukopenia, thrombocytopenia, and low C-reactive protein, with a high sensitivity of 93.1% and specificity of 96.1%. However, this score was not applicable to other febrile infectious diseases, and there is a limit to the application of only a small number of laboratory variables.

Although previous studies on the relationship between environmental factors and SFTS risk are scarce, some reports have suggested that meteorological factors such as temperature, amount of rainfall, and humidity levels were associated with SFTS infection by increasing the rate of SFTS virus (SFTSV) transmission to humans [5]. Temperatures perceived as comfortable can promote vector growth and activity, affecting human and animal behavior, which may increase the risk of SFTSV transmission [13]. A previous study reported that the optimal temperature and humidity for breeding of *Haemaphysalis longicornis,* a major vector for SFTSV transmission, were 22 ± 2 °C and 85 ± 10%, respectively. Thus, the climatic variables can also be used as predictors of infection.

This study aimed to demonstrate the role of climatic factors as a predictor of SFTS diagnosis and develop a clinical scoring system with improved accuracy that could be used in the differential diagnosis of SFTS using climate variates and clinical characteristics.

## 2. Materials and Methods

### 2.1. Patients

We compared the climatic, demographic, clinical, and laboratory characteristics of suspected and confirmed SFTS cases treated at a single tertiary hospital on Jeju Island in South Korea from April 2013 to December 2019. We included patients with suspected SFTS who were febrile, had thrombocytopenia, and were inhabitants of Jeju Island, South Korea (which had a subtropical climate and an average temperature in 2019 of 17.1 °C) [14].

We extracted data on patient sociodemographic characteristics, medical history, occupation, history of tick bites, history of contact with SFTS patients obtained from interviews with each patient, climatic characteristics (mean ambient temperature, humidity levels, and mean rainfall during the two weeks preceding the SFTS tests using Korea meteorological administration data based on the area of each patient’s residence, since the mean incubation period after SFTSV infection is 1–2 weeks) [15], and laboratory parameters, including a complete blood count and blood chemistry (obtained during the first visit to the hospital). The presence of the SFTSV was confirmed based on the detection of the S and M segment gene of the SFTSV RNA using RT-PCR tests. Except for several SFTS patients in 2013–2014, SFTS RT-PCR tests were performed on the day or after 1–2 days of hospitalization since 2015.

### 2.2. Definition and Viral Genetic Testing

The SFTS-positive group (SPG) included patients with positive SFTSV RT-PCR results while the SFTS-negative group (SNG) included patients with negative results.

Viral RNA was extracted from each patient’s first acute-phase serum using a QIAamp Viral RNA Mini kit (Qiagen Inc., Mainz, Germany) according to the manufacturer’s instructions. The extracted RNA was preserved in an elution buffer at −70 °C until RT-PCR was performed. The RT-PCR of the partial S and M segments of SFTSV was performed for molecular diagnosis [16]. The RT-PCR mixture contained 8 µL of one-step RT-PCR premix, 7 µL of detection solution, and 5 µL of the RNA template (total volume of 20 µL). The RT-PCR involved the following cycles: 30 min at 45 °C, 10 min at 90 °C, 45 cycles of 15 s at 95 °C, and 30 s at 48 °C [17]. The RT-PCR products were sequenced using a BigDye Terminator Cycle Sequencing kit (Perkin Elmer Applied Biosystems, Warrington, UK).

### 2.3. Statistical Analysis

Categorical variables were compared using the chi-square test; continuous variables were compared using the two sample *t*-test. The patients’ level of comorbidities was assessed using the Charlson Comorbidity Index Score (CCI) for each patient’s previous diagnostic disease. A multivariate logistic regression analysis was performed using the risk factors that were found to be significantly associated with SPG or SNG in the univariate analysis. Variables with *p*-values < 0.1 in the univariate analysis were included in the multivariate analysis. *p*-values < 0.05 indicated statistically significant associations in the multivariate analysis. Receiver operating characteristic (ROC) curve and area under the curve (AUC) analyses were performed to assess the predictive power of the constructed model. AUC was used as a measure of the accuracy of a predictive score. In general, an AUC of 0.5 suggested no discrimination, 0.7 to 0.8 was considered acceptable, 0.8 to 0.9 excellent, and, >0.9 outstanding. All statistical analyses were performed using SPSS version 20.0 (IBM Corp., Armonk, NY, USA).

## 3. Results

### 3.1. Baseline Characteristics in SFTS-Positive Group and SFTS-Negative Group

A total of 200 patients were included in the analysis (SPG = 62 and SNG = 138). In the SNG group, confirmed diseases were distributed as follows: 18.8% of patients had scrub typhus, 14.5% had fever of unknown origin, and 9.4% had another viral infection. The mean age of the SPG group was 59.5 ± 14.7 years, and 56.5% of the patients were men. The mean age of the SNG group was 55.3 ± 19.8 years, and 50% of the patients were men. The mean CCI in the SPG group was significantly lower than that in the SNG group (0.3 ± 0.6 vs. 0.85 ± 1.1, *p* < 0.001). The history of tick bite in the SPG group was significantly higher than that in the SNG group (50% vs. 25.4%, *p =* 0.001). There was no significant difference in the type or geographical location of exposure to SFTSV (Figure 1).

Among climatic conditions, a higher mean ambient temperature and humidity were associated with the SPG group rather than with the SNG group (22.5 ± 4.2 °C vs. 18.9 ± 5.7 °C, *p* < 0.001; 77.9 ± 7.7% vs. 70.7 ± 7.6%, *p* < 0.001, respectively) (Table 1).

At presentation, fever (*p* = 0.001), poor oral intake (*p* < 0.001), dizziness (*p* = 0.045), nausea (*p* = 0.001), diarrhea (*p* < 0.001), cough (*p* = 0.025), and lymphadenopathy (*p* < 0.001) were more common in the SPG group than in the SNG group. In both groups, febrile duration was longer in the SPG group than in the SNG group. The comparisons of laboratory findings between groups are presented in Table 2. In SPG, mean SFTS viral loads at diagnosis time were 932,183 ± 40,154,428 copies, peaked in May (mean 2,141,445 copies ± 16,091,117), and were lowest in October (mean 57,308 ± 61,524). In addition, the mortality of patients with SFTS (6/9 SFTS patients) was very high in May and June.

### 3.2. Risk Factors Associated with SFTS

In the univariate logistic regression, fever ≥ 38.0 °C (*p* < 0.001), dizziness (*p* < 0.001), poor oral intake (*p* < 0.001), lymphadenopathy (*p* < 0.001), average ambient temperature ≥ 20 °C (*p* < 0.001), humidity ≥ 74% (*p* < 0.001), amount of rainfall ≥ 10 mmL (*p* < 0.001), white blood cell ≤ 4000 cells/μL (*p* < 0.001), absolute neutrophil count (ANC) ≤ 2000 cells/μL (*p* < 0.001), lymphocyte fraction ≥ 28% (*p* = 0.01), platelet ≤ 90,000 cells/μL (*p* = 0.001), C-reactive protein (CRP) ≤ 1.2 mg/dL (*p* < 0.001), creatinine kinase (CK) ≥ 200 IU/L (*p* = 0.006), activated partial thromboplastin time (aPTT) ≥ 35 s (*p* < 0.001), and creatinine ≤ 1.8 mg/dL (*P* = 0.05) were significantly associated with the SPG group (Table 3).

In the multivariate logistic regression, poor oral intake (odds ratio [OR] = 5.87, 95% confidence interval [CI]: 2.42–8.25, *p* = 0.002), lymphadenopathy (OR = 7.20, 95% CI: 6.24–11.76 *p* < 0.001), mean ambient temperature ≥ 20 °C (OR = 4.60, 95% CI: 1.46–10.28, *P* = 0.016), neutropenia (absolute neutrophil count [ANC] ≤ 2000 cells/μL) (OR = 8.95, 95% CI: 2.30–21.25, *p* = 0.005), C-reactive protein (CRP) ≤ 1.2 mg/dL, (OR = 6.42, 95% CI 4.02–24.21, *p* < 0.001), and creatinine kinase (CK) ≥ 200 IU/L (OR = 5.94, 95% CI: 1.42–24.92, *p* = 0.015) were significantly associated with the SPG group (Table 3).

### 3.3. Clinical Prediction Score for the SFTS Clinical Model

A prediction score for the SPG group vs. the SNG group was created using a combination of the parameters identified as statistically significant in the multivariate analysis; poor oral intake, lymphadenopathy, ambient temperature ≥ 20 °C, ANC ≤ 2000 cells/μL, CRP ≤ 1.2 mg/dL, and CK 200 IU/L; an allocation of zero points if the criterion was absent and one point if the criterion was present with the total sum ranged from zero to six. The ROC curve indicated that the optimal cut-off score for the model was ≥ 3 points. A score ≥ 3 had an 82.3% sensitivity and 84.8% specificity for SPG, with a ROC AUC of 0.891 (95% CI: 0.837–0.944) (Figure 2).

Otherwise, without ambient temperature, the ROC for a model based on the remaining five factors had a cut-off value ≥ 3, 61.3% sensitivity and 94.9% specificity, with ROC AUC of 0.876.

## 4. Discussion

In the present study, we compared the characteristics of suspected and confirmed SFTS cases and found that mean ambient temperature would be a predictor of SFTS on Jeju Island. The present findings suggest that patients with fever and thrombocytopenia, including patients exposed to outdoor activities in ambient temperature ≥ 20 °C, should be referred for an SFTS RT-PCR test. Furthermore, the present study proposed a predictive score system for SFTS that is based on six parameters that can be easily established in a general clinical setting. Six parameters that predicted SFTS infection were identified from the multivariate analysis: poor oral intake, lymphadenopathy, ambient temperature ≥ 20 °C, ANC ≤ 2000 cells/μL, CRP ≤ 1.2 mg/dL, and CK levels ≥ 200 IU/L; which showed high sensitivity and specificity (both > 80%) and good validity from the AUC > 0.8.

This study compared the climatic variables of suspected and confirmed SFTS cases two weeks before the diagnosis. Data from the Korean Center for Disease Control and Prevention for 2013–2018 suggested that SFTS infections on Jeju Island occurred mainly during the spring and summer seasons; this is in contrast to some of the other regions, for example, the Kangwon-do and Gyeongbuk regions, where the incidence of SFTS is high during the summer and autumn seasons [9]. On Jeju Island during 2013–2019, the number of infected ticks tended to surge in April, while the number of new SFTS cases tended to increase sharply during May [18] (Figure 3).

Compared to the Korean Peninsula, Jeju Island maintains a temperature > 20 °C during May to October. Overall, due to its climate, Jeju Island is characterized by a higher growth rate of SFTSV-infected ticks that starts in the spring and yields a tick population greater than that found in other regions [18]. In addition, Jeju Island has a large agricultural population, which means the risk of exposure to ticks is relatively high, driving up the incidence of SFTS. This combination of factors can affect the role of temperature as an independent risk factor for SFTS transmission on Jeju Island.

SFTS symptoms reported in previous studies included fever and gastrointestinal symptoms [1,8,11], alongside regional lymphadenopathy, leukocytopenia, thrombocytopenia, and low CRP levels [8,11,12,15]. In addition, elevated liver enzyme levels, the presence of coagulation disorders, and elevated levels of CK and LDH have been reported [13,19]. High CK levels in SFTS patients have been associated a high mortality rate [8]. From the univariate analysis in the present study, fever, nausea, diarrhea, and lymphadenopathy were clinical symptoms that were found to be significantly associated with SFTS, which is consistent with findings from previous studies. Poor oral intake was another significant symptom in this study. Similarly, leukocytopenia, neutropenia, thrombocytopenia, low CRP levels, prolonged aPTT, and high CK level were significantly associated with SFTS; however, neither the liver enzymes nor the LDH levels differed between the groups.

Previous studies proposed several scoring systems to predict SFTS, including two studies conducted in Korea [11,12]. One study used three variables to generate a prediction score: leukopenia (WBC count < 4000/mm^3^), thrombocytopenia (platelet count < 80,000/mm^3^), and a low CRP level (<1 mg/dL). The resulting score ≥ 2 had a sensitivity of 93.1% and a specificity of 96.1% for SFTS. A separate study developed an SFTS differentiation score based on four parameters; altered mental status, leukopenia, prolonged aPTT, and a normal CRP level. In this study, the optimal cut-off score was > 1 and had 100% sensitivity and 97% specificity for SFTS. However, these previous studies compared SFTS with scrub typhus rather than other diseases with similar clinical presentations. In contrast, our study aimed to develop a scoring system that could differentiate SFTS from similar diseases in clinical settings, where SFTS RT-PCR tests were performed.

Compared with other predictive scoring systems including the two above-mentioned studies comparing two SFTS and scrub typhus patients [11,12,20], leukopenia, thrombocytopenia, and a normal or low CRP level were used as clinical parameters, in the present model, we included a low CRP level and neutropenia (instead of leukopenia). A variable for levels of thrombocytopenia was not included in the model due to a non-significant association in multivariate analysis (this finding was likely due to the tendency to refer suspected SFTS cases that present with thrombocytopenia for RT-PCR testing). The previously proposed predictive scoring systems included clinical and laboratory parameters of SFTS. However, in the present study, the proposed predictive scoring system included ambient temperature as one of the parameters. Without ambient temperature, the ROC for a model based on the remaining five factors had a cut-off value ≥ 3, sensitivity of 61.3%, and specificity of 94.9%, with sensitivity significantly reduced compared to that of the complete model (82.3%); the AUC value also decreased from 0.891 to 0.876 after ambient temperature was removed from the model. These findings suggest that ambient temperature may be useful in the clinical assessment of suspected SFTS cases.

This study has several limitations. First, this was a single-center study influenced by the climatic characteristics of Jeju Island. In addition, our data did not include a comparison of the clinical characteristics of patients with SFTS and ticks between Jeju Island and mainland, South Korea. However, Jeju Island is an endemic area with the highest annual incidence of SFTS in South Korea and given the large proportion of the total number of Korean SFTS cases that occur on Jeju Island, the presented findings are likely to be representative of the entire population. Second, the present study was retrospective, which may have produced results with a selection bias. Third, all the parameters included in the scoring system were assigned one point each, irrespective of their relative incidence. Therefore, to develop a more accurate predictive scoring system, further research that calculates the relative incidence of parameters and reflects them in the weight of the score, will be important. Fourth, the ambient temperature could not distinguish between SFTS and other tick-borne diseases. However, this meteorological factor may be an application with specific findings of SFTS. Finally, cases of false-negative SFTS may have been included in the present study sample since the SNG group was not tested for antibodies against SFTSV.

## 5. Conclusions

This study in examining climate conditions as relevant factors in the clinical assessment of suspected SFTS cases showed that ambient temperature was significantly associated with SFTS, suggesting it would be an independent diagnostic factor. This study has proposed a simple scoring system, using climate and clinical parameters to help assess the risk of SFTS in patients presenting at hospital. The system is based on six parameters that can be easily assessed in primary care settings, enabling the decision for early SFTS testing.

## Figures and Tables

**Figure 1 viruses-13-00010-f001:**
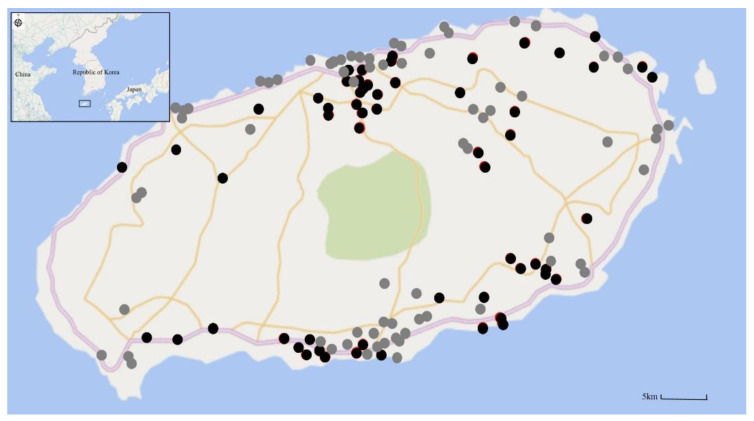
Geographic distribution of patients of the severe fever with thrombocytopenia syndrome (SFTS) and non-SFTS groups on Jeju Island (2013–2019). Black closed circles indicate the site of SFTS virus (SFTSV) infection, and gray closed circles indicate the site of non-SFTS disease.

**Figure 2 viruses-13-00010-f002:**
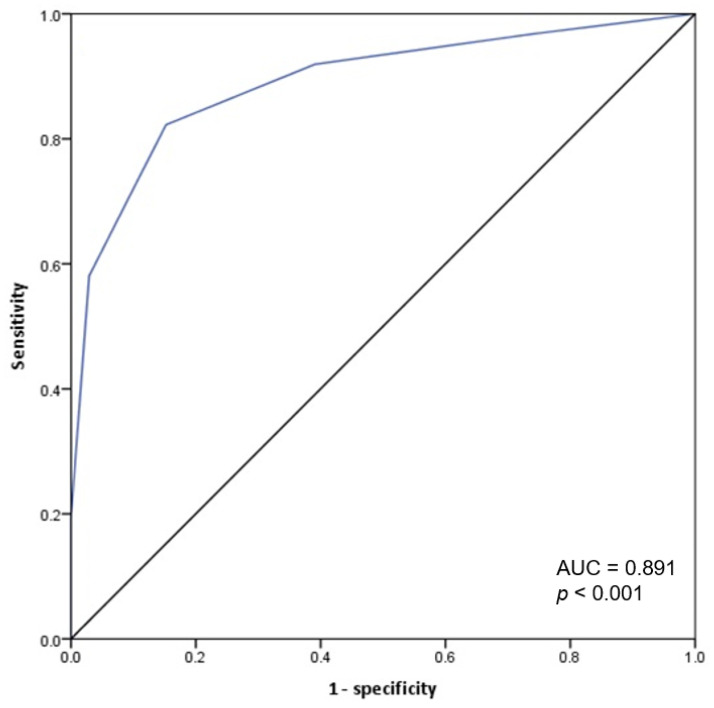
Receiver operating characteristics (ROC) curve for the predictive score of SFTS group. Prediction score system = (1 × poor oral intake) + (1 × lymphadenopathy) + (1 × ambient temperature ≥ 20 °C) + (1 × ANC ≤ 2000 cells/μL) + (1 × CRP ≤ 1.2 mg/dL) + (1 × CK ≥ 200 IU/L).

**Figure 3 viruses-13-00010-f003:**
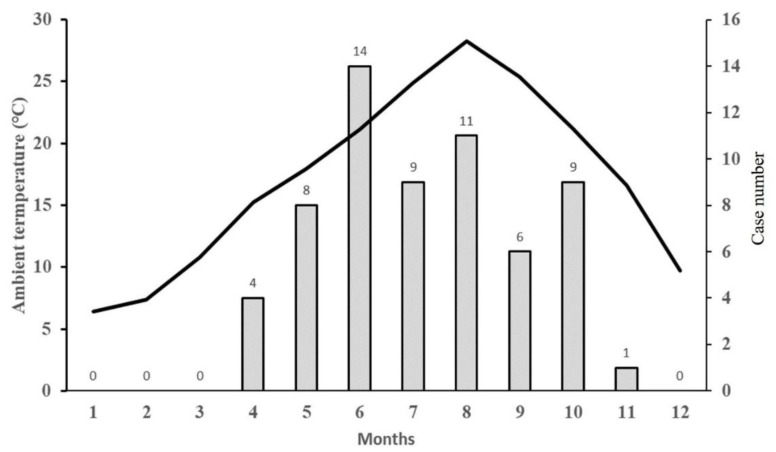
Relationship between monthly mean ambient temperature and incidence of patient with severe fever with thrombocytopenia syndrome. The black line indicates the mean monthly ambient temperature on Jeju Island (2013–2019). The gray bars indicate the monthly total incidence cases of SFTS in patients on Jeju Island (2013–2019).

**Table 1 viruses-13-00010-t001:** Baseline demographic and clinical characteristics between patients with positive (SPG) and negative (SNG) severe fever with thrombocytopenia syndrome reverse transcription polymerase chain reaction (RT-PCR).

Variable	SPG (*n* = 62)	SNG (*n* = 138)	*p*-Value
Age (years)	59.5 (±14.7)	55.3 (±19.8)	0.102
Male gender	35 (56.5)	69 (50)	0.398
CCI	0.3 (±0.6)	0.85 (±1.1)	<0.001
SFTS viral load, median (IQR)	57,121 (3492–267,966)	N/A	
Exposure type			<0.001
Agriculture	24 (38.7)	26 (18.8)	
Livestock	7 (11.3)	5 (3.6)	
Leisure activity	8 (12.9)	15 (10.9)	
Grave	7 (11.3)	2 (1.4)	
Gathering	7 (11.3)	7 (5.1)	
Patient contact	5 (8.1)	9 (6.5)	
Other	4 (6.5)	74 (38.4)	
Tick bite	31(50.0)	35 (25.4)	0.001
Geographical location			0.105
Northern	12 (19.4)	44 (31.9)	
North-eastern	15 (24.2)	23 (16.7)	
North-western	11(17.7)	14 (10.1)	
South	5 (8.1)	16 (11.6)	
South-eastern	12 (19.4)	21 (15.2)	
South-western	7 (11.3)	12 (8.7)	
Other	0 (0)	8 (5.8)	
Seasonal occurrence			<0.001
Spring	9 (14.5)	16 (11.6)	
Summer	37 (59.7)	44 (31.9)	
Autumn	16 (25.8)	63 (45.7)	
Winter	0 (0)	15 (10.9)	
Climatic conditions			
Ambient temperature (°C)	22.5 (±4.2)	18.9 (±5.7)	<0.001
Humidity, (%)	77.9 (±7.7)	70.7 (±7.6)	<0.001
Rainfall amount, (mmL)	12.0 (±14.3)	8.7 (±11.9)	0.091
Duration			
From onset of illness to admission	3.9 (±2.2)	4.9 (±6.4)	0.129
30-days mortality (%)	11.3	6.5	0.339
All-cause Mortality (%)	11.3	8.0	0.448

*p* values are derived from a two sample *t*-test for continuous data and a chi-square test for categorical data. Continuous data are presented as means and standard deviations; categorical data are presented as counts and percentages. Abbreviation: CCI = comorbidity index score (calculated by the Charlson Comorbidity Index), IQR = interquartile range, N/A = not available.

**Table 2 viruses-13-00010-t002:** Clinical symptoms and laboratory findings at initial presentation for patients with positive (SPG) and negative (SNG) results severe fever with thrombocytopenia syndrome using RT-PCR.

Variable	SPG (*n* = 62)	SNG (*n* = 138)	*p*-Value
Fever (n)	55 (90.2)	89 (69.0)	0.001
Febrile duration (day)	4.7 (±3.5)	3.3 (±3.7)	0.031
Fever peak (°C)	38.5 (±0.8)	38.4 (±1.5)	0.356
Chill	26 (42.6)	48 (37.2)	0.475
Fatigue	12 (19.7)	33 (25.6)	0.371
Headache	17 (27.9)	27 (20.9)	0.290
Myalgia	26 (42.6)	44 (34.1)	0.256
Dizziness	12 (19.7)	12 (9.3)	0.045
Poor oral intake	25 (41.0)	10 (7.8)	<0.001
Nausea	18 (29.5)	14 (10.9)	0.001
Vomiting	7 (11.5)	12 (9.3)	0.641
Abdominal pain	4 (6.6)	18 (14.0)	0.137
Diarrhea	25 (41.0)	16 (12.4)	<0.001
Hematochezia	0 (0.0)	2 (1.6)	0.332
Hematuria	0 (0.0)	8 (6.2)	0.047
Cough	2 (3.3)	18 (14.0)	0.025
Sputum	2 (3.3)	10 (7.8)	0.237
Hemoptysis	2 (3.3)	1 (0.8)	0.196
Dyspnea	2 (3.3)	7 (5.4)	0.515
Convulsion Altered mentality	0 (0.0) 11 (17.7)	0 (0.0) 10 (7.8)	0.041
Lymphadenopathy	16 (25.8)	1 (0.8)	<0.001
Hemorrhagic plaque	0 (0.0)	3 (2.3)	0.226
WBC (cells/μL)	2554 (±2235)	6659 (±5494)	<0.001
ANC (cells/μL)	1593 (±1876)	4757 (±5010)	<0.001
Lymphocyte fraction (%)	36.1 (±16.4)	25.5 (±15.8)	<0.001
Hb (g/dL)	14.0 (±2.1)	13.33 (±3.5)	0.130
Hematocrit (%)	39.9 (±6.5)	38.1 (±6.4)	0.084
Platelet (cells/μL)	91982 (±42,698)	125050 (±81,418)	<0.001
CRP (mg/dL)	1.2 (±2.70)	9.2 (±9.94)	<0.001
PT (INR)	1.25 (±1.35)	1.92 (±8.75)	0.574
aPTT (sec)	41.8 (±12.5)	35.3 (±11.3)	0.001
Albumin (g/dL)	3.7 (±0.5)	3.6 (±0.7)	0.243
AST (IU/L)	167 (±292)	237 (±947)	0.595
ALT (IU/L)	72 (±85)	162 (±680)	0.321
LDH (IU/L)	1022 (±1274)	1066 (±1541)	0.856
CK (IU/L)	1062 (±1736)	431 (±821)	0.014
BUN (mg/dL)	17.5 (±18.8)	21.3 (±18.9)	0.195
Creatinine (mg/L)	1.1 (±0.6)	1.5 (±1.3)	0.025
MODS	2.4 (±2.5)	2.7 (±2.8)	0.463

*p* values are derived from a two sample *t*-test for continuous data and chi-square test for categorical data. Continuous data are presented as means and standard deviations; categorical data are presented as counts and percentages. Abbreviation: WBC = white blood cell, ANC = absolute neutrophil count, Hb = hemoglobin, CRP = C-reactive protein, PT = prothrombin time, INR = international normalized ratio, aPTT = activated partial thromboplastin time, AST = aspartate aminotransferase, ALT = alanine aminotransferase, LDH = lactate dehydrogenase, CK = creatinine kinase, BUN = blood urea nitrogen, MODS = multiple organ dysfunction score.

**Table 3 viruses-13-00010-t003:** Univariate and multivariate logistic regression analyses of factors associated with positive results for severe fever with thrombocytopenia syndrome using RT-PCR.

Variable		Univariate			Multivariate	
	OR	CI (95%)	*p*-Value	OR	CI (95%)	*p*-Value
Fever ≥ 38.0 °C	4.15	1.64–10.35	<0.001			
Dizziness	3.42	1.03–5.69	<0.001			
Poor oral intake	5.81	3.60–7.66	<0.001	5.87	2.42–8.25	0.002
Diarrhea	4.32	2.36–10.19	0.319			
Lymphadenopathy	6.36	5.74–10.21	<0.001	7.20	6.24–11.76	<0.001
Ambient temperature ≥ 20 °C	4.18	2.19–8.57	<0.001	4.62	1.46–10.28	0.016
Humidity ≥ 74%	4.17	2.72–10.51	<0.001			
Rainfall amount ≥ 10 mmL	2.34	1.19–4.16	<0.001			
WBC ≤ 4000 cells/μL	14.96	6.00–37.33	<0.001			
ANC ≤ 2000 cells/μL	15.70	6.78–36.36	<0.001	8.95	2.30–21.25	0.005
Lymphocytes fraction ≥ 28%	2.93	1.53–5.59	0.01			
Platelet ≤ 90,000 cells/μL	2.80	1.51–5.21	0.001			
CRP ≤ 1.2 mg/dL	7.81	5.61–21.63	<0.001	6.42	4.02–24.21	<0.001
CK ≥ 200 IU/L	2.63	1.32–5.23	0.006	5.94	1.42–24.92	0.015
aPTT ≥ 35 sec	4.74	2.32–9.68	<0.001			
Creatinine ≤ 1.8 mg/dL	2.75	1.00–7.57	0.05			

Abbreviation: WBC = white blood cell, ANC = absolute neutrophil count, CRP = C-reactive protein, CK = creatinine kinase, aPTT = activated partial thromboplastin time.

## Data Availability

The data are not publicly available due to privacy of the patients (eg address).

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
