# Peer review of "Prognostic Factors of Severe Fever with Thrombocytopenia Syndrome in South Korea"

_viruses, 2020, doi:10.3390/v13010010_

Round 1

Reviewer 1 Report

In this manuscript, the authors analyzed a total of 200 patients (suspected and confirmed SFTS cases) treated at a single tertiary hospital in South Korea from April 2013 to December 2019 and found an association of climatic factor (higher ambient temperature) and several clinical parameters (poor oral intake, lymphadenopathy, ANC<2000 cells/mL, CRP<1.2 mg/dL, CK levels>200 IU/L) with the SFTS-positive group. The following issues need to be addressed to improve this manuscript before publication.

  1. In the Discussion section, the authors conclude that "overall, due to its climate, Jeju Island is characterized by a higher growth rate of SFTSV-infected ticks that starts in the spring and yields a tick population greater than that in other regions" (lines 197-199). However, this is not the conclusion based on this study's data since no tick survey data is available in the present study. If the authors aim to characterize the reason for SFTS endemicity in Jeju Island based on its climatic characteristic and tick population, data of "climate", "SFTSV prevalence in ticks", and "SFTS cases in humans" need to be collected from both Jeju Island and other regions (e.g., mainland) and be compared. Alternatively, if the authors aim to demonstrate a seasonal change in SFTS cases in humans with ambient temperature correlation, it would be better to focus on a certain region (e.g., Jeju Island). As the authors state, Jeju Island's average annual temperature is higher than that in the mainland. If SFTSV prevalence in ticks is higher in Jeju Island than the mainland, this would be a confounding factor to bias the correlation between temperature and SFTS cases. Thus, it is not appropriate to combine data obtained from Jeju Island and the mainland.

  1. The authors state that "the present findings suggest that patients with fever and thrombocytopenia, including patients exposed to outdoor activities in ambient temperature>20°C, should be referred for an SFTS RT-PCR test" (lines174-176). However, this needs to be toned down because standard deviations of ambient temperature for both SFTSV-positive and -negative groups are large (22.5±4.2 and 18.9±5.7, respectively) (Table 1).

  1. The authors state that "mean ambient temperature can be a predictor of SFTS" (line 174) and "ambient temperature may be useful in the clinical assessment of suspected SFTS cases" (lines 235-236). However, it is little questionable whether ambient temperature can be used as a predictor of SFTS specifically or tick-borne diseases generally. Did the authors find any patients with other tick-borne diseases in their analysis? In that case, the ambient temperature could distinguish SFTS and other tick-borne diseases?

  1. The authors state that climatic characteristics were extracted during the two weeks preceding the SFTS tests using Korea meteorological administration data (lines 71-72). Was the RT-PCR test always performed on the day the patient visited the hospital? Were those climatic characteristics used in the analysis obtained from each area of patient residence? These details need to be clarified.

  1. The resolution of Figure 3 is low. This figure needs to be replaced with a new one with better resolution.

  1. The y-axis of Figure 3 indicates ambient temperature and SFTS case number, and the x-axis indicates the month from January to December. Is this figure generated using data harvested within a year or data of average ambient temperature and cumulative SFTS cases from 2013 to 2019? This needs to be explained in the graph.

  1. The authors provide Figure 3a and 3b. These figures need to be referred to in the manuscript separately, instead of using the term "Figure 3" (line 190).

  1. The term "Phlebovirus" is no longer used (line 30). SFTSV is currently classified in the genus Banyangvirusin the family Phenuiviridae of the order Bunyavirales.

Author Response

Thank you for reviewing our paper.

Manuscript ID: viruses-1023201
Type of manuscript: Article
Title: Prognostic factors of severe fever with thrombocytopenia syndrome in Jeju Island, South Korea

Reviewer 1

In this manuscript, the authors analyzed a total of 200 patients (suspected and confirmed SFTS cases) treated at a single tertiary hospital in South Korea from April 2013 to December 2019 and found an association of climatic factor (higher ambient temperature) and several clinical parameters (poor oral intake, lymphadenopathy, ANC<2000 cells/mL, CRP<1.2 mg/dL, CK levels>200 IU/L) with the SFTS-positive group. The following issues need to be addressed to improve this manuscript before publication.

  1. In the Discussion section, the authors conclude that "overall, due to its climate, Jeju Island is characterized by a higher growth rate of SFTSV-infected ticks that starts in the spring and yields a tick population greater than that in other regions" (lines 197-199). However, this is not the conclusion based on this study's data since no tick survey data is available in the present study. If the authors aim to characterize the reason for SFTS endemicity in Jeju Island based on its climatic characteristic and tick population, data of "climate", "SFTSV prevalence in ticks", and "SFTS cases in humans" need to be collected from both Jeju Island and other regions (e.g., mainland) and be compared. Alternatively, if the authors aim to demonstrate a seasonal change in SFTS cases in humans with ambient temperature correlation, it would be better to focus on a certain region (e.g., Jeju Island). As the authors state, Jeju Island's average annual temperature is higher than that in the mainland. If SFTSV prevalence in ticks is higher in Jeju Island than the mainland, this would be a confounding factor to bias the correlation between temperature and SFTS cases. Thus, it is not appropriate to combine data obtained from Jeju Island and the mainland.

: Thank you for your opinion, and we agreed all about your comments.

 Based on a tick data in Jeju Island (reference [18] Yoo et al. Emerg Infect Dis. 2020 Sep; 26 (9): 2292-2294, this author is this study’s correspondence) which patients with SFTS in Jeju Island is more early reported by Korean CDC, and it is characterized by a higher growth rate of SFTSV-infected ticks that starts from the spring in Jeju Island than that in mainland, we added a reference to this sentence. In addition, study of this data was almost simultaneously proceeded our previous study. However, our data did not compare between Jeju Island and the mainland, South Korea, because of other studies of the mainland were collected samples as an extra sample for another disease in the hospital, and retrospective study, So, we descripted the limitation of our study in the limitation section.

And, in the discussion, we tried to describe that Jeju Island maintains a temperature of >20°C from May to October, therefore the amount of SFTSV-infected ticks increases from spring compared to the mainland. We revised the sentence more clearly, and we were tone down about climatic characteristics in Jeju Island as your comment

  1. The authors state that "the present findings suggest that patients with fever and thrombocytopenia, including patients exposed to outdoor activities in ambient temperature>20°C, should be referred for an SFTS RT-PCR test" (lines174-176). However, this needs to be toned down because standard deviations of ambient temperature for both SFTSV-positive and -negative groups are large (22.5±4.2 and 18.9±5.7, respectively) (Table 1).

: Thank you for your kind opinion, we are all agreed.

However, difference of mean ambient temperature between both two groups were significant as P < 0.001 in T-test. In addition, “ambient temperature ≥20°C” was significant in univariate and multivariate analyses. Mean monthly ambient temperature ≥ 20°C was from June to October in Figure 3, but if a patient had at our suggested factors such as period of mean weakly ambient temperature ≥ 20°C in April, May, and November, fever, thrombocytopenia, low CRP, and lymphadenopathy, we would be a recommendation of SFTS RT PCR. So, we are tone down a description from “can” to a “would” as your comments.

  1. The authors state that "mean ambient temperature can be a predictor of SFTS" (line 174) and "ambient temperature may be useful in the clinical assessment of suspected SFTS cases" (lines 235-236). However, it is little questionable whether ambient temperature can be used as a predictor of SFTS specifically or tick-borne diseases generally. Did the authors find any patients with other tick-borne diseases in their analysis? In that case, the ambient temperature could distinguish SFTS and other tick-borne diseases

: In this study, in SPG and SNG patients in Jeju, there was a statistically significant difference in mean ambient temperature (especially ≥20°C) for an average of 2 weeks at the time of diagnosis. When multivariate analysis 6 parameters which including the mean ambient temperature ≥20°C, the sensitivity and AUC values ​​are higher compared with result of multivariate analysis with 5 parameters without ambient temperature. Also, there was a positive correlation between ambient temperature, SFTS-infected tick incidence and the number of patients in Jeju Island. Based on this result, it was determined that the average temperature can be considered as a predictive factor for SFTS.

Therefore, when we suspect SFTSV infection through fever, thrombocytopenia and other laboratory findings, we can consider the progress of the test by considering the average temperature at the time when performing SFTS PCR (e.g., SFTS in late spring and summer when the average temperature is higher than 20°C. It was determined that it could be used as a factor).

In addition, 18.8% of SNG (SFTS negative group-patients with clinically suspected SFTS but actually negative for SFTS PCR) where scrub typhus, which is another vector-borne disease in this study. Scrub typhus was developed in Jeju Island from October to December. So, there is possible of distinction some degree between SFTS and scrub typhus as an occurrence time.

However, we didn’t analyze the relationship of ambient temperature between SFTS patients and the subgroup of only other tick borne diseases among SNG, so it can be considered as a limitation of another study. There are ticks such as about 86% of Haemaphysalis longicornis and 4.0 % of Ixodex spp. in South Korea. These ticks causes diseases of SFTS, Theileria, Babesia, Anaplasma, Bartonella, Q fever, and Ehrilichia in South Korea. Theses disease very rarely occur in South Korea based on data of KCDC except Q fever. For this reason, it is assumed that the probability of occurrence is low. In the future, we will introduce a study of another tick borne disease including Q fever in Jeju Island and analyze the correlation between the ambient temperature between SFTS and other tick borne diseases.

  1. The authors state that climatic characteristics were extracted during the two weeks preceding the SFTS tests using Korea meteorological administration data (lines 71-72). Was the RT-PCR test always performed on the day the patient visited the hospital? Were those climatic characteristics used in the analysis obtained from each area of patient residence? These details need to be clarified.

: Except for SFTS patients in Jeju Island in 2013-2014, RT-PCR tests were performed on the day of hospitalization or within 1-2 days. In addition, the patient's residence in Jeju Island and the estimated infected area were investigated to obtain and analyze the climatic characteristics of the area. We have corrected this clearly in the method section.

  1. The resolution of Figure 3 is low. This figure needs to be replaced with a new one with better resolution.

: The figure was corrected it in high resolution.

  1. The y-axis of Figure 3 indicates ambient temperature and SFTS case number, and the x-axis indicates the month from January to December. Is this figure generated using data harvested within a year or data of average ambient temperature and cumulative SFTS cases from 2013 to 2019? This needs to be explained in the graph.

: Thank you for your opinion. It was the average value from 2013 to 2019, and it is not cumulated cases. We descripted a detailed explanation, it in the figure legend.

  1. The authors provide Figure 3a and 3b. These figures need to be referred to in the manuscript separately, instead of using the term "Figure 3" (line 190).

- Thank you for your opinion, we corrected. And, we deleted the Figure 3B as reviewer 2 comments. So, we described a part of SFTS viral load in the Result sections.

  1. The term "Phlebovirus" is no longer used (line 30). SFTSV is currently classified in the genus Banyangvirus in the family Phenuiviridae of the order Bunyavirales.

: Thank you for your opinion, we corrected.

Reviewer 2 Report

This manuscript (Kim et al.) provides a retrospective study to attempt delineation of clinical and environmental correlates of SFTS in human patients. The study is based on comparison of groups of SFTSV-positive and negative patient from Jeju Island, which has a high incidence of SFTS, using a large number of clinical parameters. Based on their analysis, the authors propose using six indicators that might be useful as an initial screen by physicians for recommendation of more definitive SFTS via virus-specific RT-PCR. These include four measures from standard blood analysis, along with “poor oral intake” and increased ambient temperature.

 The univariate and multivariate analysis appears to have been carried out reasonably, and the authors nicely point out potential limitations of their analysis in the discussion. The manuscript provides some information that might be of diagnostic use in clinical setting where access to RT-PCR for SFTSV is limited. However, I think the overall impact of study is limited.

Additional comments:

  1. Line 30-31. The bunyavirus nomenclature changed a few years ago. It would good to indicate the virus is a member of the Phenuivirus family and the Bunavirales Order.
  2. While high ambient temp correlates with higher probability of SFTS this is not really surprising since it’s already known that SFTS is more prevalent in warmer temperatures. This is presumably due to increased tick activity and corresponding increased transmission to humans.
  3. In Fig. 3b, it’s difficult to see the SFTS RNA units. However, I’m skeptical about the conclusion that the initial virus load from patients is higher when ambient temp is higher.
    1. Is this a linear or log scale? If linear, the monthly differences aren’t actually very dramatic and could be due to variables other than temperature.
    2. In May, when virus load is highest, how many patients are included in the calculation? It would be potentially interesting, and important, to see values from individual patients within this month.
    3. What is called “initial virus load” is presumably the first RT-PCR result following possible symptoms. However, since it’s highly unlikely that patient samples were obtained at identical times virtually immediately after infection, it’s likely that virus replication has occurred through different numbers of cycles in different patients.

Author Response

Thank you for reviewing our paper.
I agree with all of your comments, and the responses to the comments are attached as a word file, and some contents have been revised according to the comments.

Round 2

Reviewer 1 Report

The authors properly responded all questions this reviewer asked.